Morphological traits: predictable responses to macrohabitats across a 300 km scale

Yates Michelle L. 1
Andrew Nigel R. 1 nigel.andrew@une.edu.au
Binns Matthew 1
Gibb Heloise 2
1 Centre for Behavioural and Physiological Ecology, Zoology, University of New England , Armidale, NSW , Australia
2 Department of Zoology, La Trobe University , Melbourne, VIC , Australia
Higley Leon
Electronic publication date: 2014 Mar 4
Publication date: 2014
Volume: 2
Electronic Location ID: e271
Received 2013 Oct 27; Accepted 2014 Jan 22
Copyright: © 2014 Yates et al.
Copyright year: 2014
Copyright holder: Yates et al.
License: This is an open access article distributed under the terms of the Creative Commons Attribution License, which permits unrestricted use, distribution, reproduction and adaptation in any medium and for any purpose provided that it is properly attributed. For attribution, the original author(s), title, publication source (PeerJ) and either DOI or URL of the article must be cited.
License URL: https://creativecommons.org/licenses/by/4.0/

Keywords: Land management, Biogeography, Functional traits, Community structure, Body size

Funding: Australian Research Council (ARC) Discovery grant DP0769961 ARC Discovery Grant DP0985886 The University of New England provided MLY with a student scholarship. NRA received an Australian Research Council (ARC) Discovery grant (DP0769961) and NRA and HG received an ARC Discovery Grant (DP0985886). The funders had no role in study design, data collection and analysis, decision to publish, or preparation of the manuscript.

==============================
Species traits may provide a short-cut to predicting generalities in species turnover in response to environmental change, particularly for poorly known taxa. We ask if morphological traits of assemblages respond predictably to macrohabitats across a large scale. Ant assemblages were collected at nine paired pasture and remnant sites from within three areas along a 300 km distance. We measured ten functional morphological traits for replicate individuals of each species. We used a fourth corner model to test associations between microhabitat variables, macrohabitats (pastures and remnants) and traits. In addition, we tested the phylogenetic independence of traits, to determine if responses were likely to be due to filtering by morphology or phylogeny. Nine of ten traits were predicted by macrohabitat and the majority of these traits were independent of phylogeny. Surprisingly, microhabitat variables were not associated with morphological traits. Traits which were associated with macrohabitats were involved in locomotion, feeding behaviour and sensory ability. Ants in remnants had more maxillary palp segments, longer scapes and wider eyes, while having shorter femurs, smaller apical mandibular teeth and shorter Weber’s lengths. A clear relationship between traits and macrohabitats across a large scale suggests that species are filtered by coarse environmental differences. In contrast to the findings of previous studies, fine-scale filtering of morphological traits was not apparent. If such generalities in morphological trait responses to habitat hold across even larger scales, traits may prove critical in predicting the response of species assemblages to global change.

Introduction

One consequence of the vast diversity of species is that a large proportion of species is not yet described taxonomically (Cardoso et al., 2011) or ecologically (Thomas, 1990). This is particularly problematic for the 5–15 million species of arthropods that make up the bulk of animal diversity (Ødegaard et al., 2000), and which are known to play critical functional roles in ecosystems (Losey & Vaughan, 2006). Recent work suggests that our understanding of ecological communities may progress most rapidly through a focus on the functional traits of organisms (McGill et al., 2006; Bihn, Gebauer & Brandl, 2010). Such an approach may be particularly pertinent for poorly known taxa, for which functional traits have the potential to offer a general understanding of species turnover in response to environmental change (Gibb & Parr, 2013). Therefore, understanding the function of species through a trait method may advance ecological knowledge.

Species possess a large range of functional traits, including behavioural, genetic, phenological, dispersal, physiological, and life history traits (Violle et al., 2007; Andrew et al., 2013a). Morphology includes some of the most accessible and functionally important traits, with the potential to be measured for significant numbers of undescribed and poorly known species. Previous studies indicate that relationships between morphology and function are universal. For example, body size scales with temperature (Bergmann, 1847; Chown & Gaston, 2010), while wing morphology predicts dispersal ability (Angelo & Slansky Jr, 1984; Norberg, 1990). Traits allow us to use convergent evolution of species occupying similar niches to determine the similarity of ecosystems with little or no taxonomic overlap (McGill et al., 2006). The trait focused approach thus allows potential for an improved understanding of broad-scale patterns and to assess impacts of environmental change (Andrew et al., 2013b).

Traits have been used to understand the structure of plant (Grime, 1977), vertebrate (Ricklefs, Cochran & Pianka, 1981), and microbial communities (Green, Bohannan & Whitaker, 2008), but their use in entomology is still in its infancy (but see Bihn, Gebauer & Brandl, 2010; Gibb & Parr, 2010; Silva & Brandão, 2010; Barton et al., 2011; Yates & Andrew, 2011, for exceptions). Here, we use ants (Hymenoptera: Formicidae) as a model taxon to explore relationships between morphology and the environment at the scale of local assemblages. Ants were selected because they: comprise the dominant fraction of animal biomass in most terrestrial communities (Wilson & Holldobler, 2005); perform a range of important ecosystem functions (Folgarait, 1998); are abundant, so likely to be trapped if present; and because new research shows strong relationships between functionally important morphological traits of ants, habitat complexity and disturbance (Bihn, Gebauer & Brandl, 2010; Chown & Gaston, 2010; Silva & Brandão, 2010).

Here, we examine the relationship between morphology, microhabitat, and macrohabitat within three areas spanning a large 300 km scale. In a previous study (Yates, Gibb & Andrew, 2012), we showed that species assemblages respond strongly to macrohabitat as well as to a range of microhabitat variables. We now extend this research by firstly characterising the assemblage morphospace and then by assessing trait responses to the environment using a new fourth corner model (Brown et al., 2014). Specifically, we asked the following questions: (1) Do morphological traits respond to the environment at macro or micro habitat scales? (2) Are the patterns detected in morphological traits independent of phylogenetic relationships? (3) Do traits related to the environment vary more within complex macrohabitats?

Materials and Methods

To test the relationship between macrohabitat, microhabitat and morphology, we selected three replicate paired sites (pasture vs remnant vegetation) in three areas (3∗2∗3 = 18 sites). The areas were Boomi (mean maximum temperature = 27.6°C; mean rainfall = 505.9 mm year; elevation = 160 m a.s.l.), Inverell (mean maximum temperature = 23.9°C; mean rainfall = 805.3 mm year; elevation = 582 m a.s.l.), and Armidale (mean maximum temperature = 20.3°C; mean rainfall = 791.5 mm year; elevation = 980 m a.s.l.), in the north east of New South Wales, Australia (BOM 2012) (see Fig. S1). We selected two very different macrohabitats across three areas which were similar in biotic and abiotic characteristics as trait responses to micro and macro habitats was of primary interest.

Remnant and pasture sites were paired to minimize differences in temperature, soil type or elevation. Remnant habitats were characterized by >80% tree cover and served primarily as conservation areas. Pasture habitats were characterized by >80% grass cover and <20% tree cover. All pastures were actively grazed by cattle, sheep, or horses. There was some leaf litter on the ground in pastures, mainly from decaying grass. Pastures had been free of tilling for the past 10 years and had experienced medium to intense grazing (medium = two livestock/hectare to intense = six livestock/hectare). Selected habitats covered a minimum of one hectare.

Microhabitat variables

Within each site, we established plots at three haphazard locations, separated by 20 m (i.e., a total of 54 plots). At the plot scale, we measured a variety of microhabitat variables, selected based on associations with ant assemblages in previous studies (Bestelmeyer & Wiens, 2001; Arnan, Rodrigo & Retana, 2006; van Ingen, Campos & Andersen, 2008). Ambient temperatures were measured within each plot, every 30 min for the entire seven day collection period. One miniature temperature logger (Thermochron i-button; Maxim Integrated Products, Sunnyvale, CA, USA) was tied to the base of the plastic cover frame of one pitfall trap in each plot (described below). All temperature loggers were in direct contact with soil. One visual assessment per plot was conducted to quantify the percentage cover of short grass, tall grass, herbs, leaf litter, canopy, bare ground and rocks as well as slope and aspect.

One soil sample was collected from each plot using an auger (approximately 20 cm depth), and was dried at room temperature for approximately eight weeks (the three plot soil samples were later pooled for each site). Samples were sieved using a 0.05 mm aperture sieve. Phosphorus (P) and Sulfur (S) were analysed using an Inductively Coupled Plasma Optical Emission Spectrometer. Carbon (C) and Nitrogen (N) were analysed using a Carlo Erba (Strada Rivoltana, Milan, Italy ) NA 1500 Solid Sample Analyser coupled to a Tracer Mass Stable Isotope Analyser (Europa Scientific, United Kingdom).

Once all environmental data was collected and collated, correlations between each variable were calculated using PC-ORD (McCune & Mefford, 2011). Correlations of r ≥ 0.8 were considered meaningful (Gibb & Parr, 2013). The microhabitat variables which were used in analyses were tall grass, short grass, herb, leaf and bareground cover, C:N, P and average ambient daily temperature (°C). Other variables that were highly correlated with those mentioned above were removed.

Ant sampling

Within each site, pitfall traps were set up in sets of five at the three haphazard plots, separated by 20 m. We used 200 ml plastic vials, 7.5 cm in diameter, filled with 80% glycol: 20% tap water, with 20 cm plastic covers to reduce evaporation and rainfall overflow. Pitfall traps were placed in an X design at each location, with a central pitfall trap surrounded by 4 traps, each being 2 m from the central trap. Pitfall traps were placed at least 20 m from habitat boundaries to avoid edge effects and were left for seven days. Due to the distances between sites (approximately 300 km between two furthest sites), traps were out for the collecting period between the 3rd and 13th November, 2007. Ant specimens were sorted to genus using Australian Ants Online (http://anic.ento.csiro.au/ants/) and identifications were verified by Mr. Steve Tremont. Specimens were then identified to ‘morphospecies’ (Oliver & Beattie, 1993). Results obtained using the morphospecies method are largely consistent with those obtained using species-level identification (Oliver & Beattie, 1996; Derraik et al., 2002). In addition, because morphospecies are based on morphological characteristics, the use of this classification may have allowed us to better account for differences in morphology within polymorphic species than would a species-based approach. The morphospecies approach did not affect our ability to detect phylogenetic dependence of traits because the ant phylogeny was only available at genus level and our genus-level identifications were accurate.

Morphological trait measurements

We commenced by measuring 20 morphological traits on a single specimen of 24 different genera (verified by HG). This represented nearly 70% of the genera collected (24 out of 36 genera). Traits in the initial set of 20 morphospecies (detailed in Table 1) were selected because they were expected to relate to the ecological role of each morphospecies. Measurements were made using a dissecting microscope designed specifically for trait measurements which used a Nikon X-Y stage micrometer wired to a SC-112 digital readout.

Table 1 The 20 morphological traits examined and the abbreviations given to these traits.

The ten traits which were selected after the test for correlations (Table S1) appear in bold. All lengths were measured in millimetres.

Morphological trait	Abbrev.	Hypothesised functional significance	
Continuous measures			
Weber’s length	Weber’s	Indicative of body size (Weber, 1938), correlates to metabolic function and habitat complexity	
Minimum inter-eye distance	MII	May be a function of the habitat in which the ant lives: having eyes further apart may be beneficial in more complex habitats (Gibb & Parr, 2013)	
Eye width	EW	Ability to see laterally (Baker, Meese & Georgeson, 2007)	
Eye length	EL	Indicates feeding behaviour; predatory ants have smaller eyes (Weiser & Kaspari, 2006)	
Head length	HL	May be indicative of diet; longer head length may indicate herbivory.	
Head Width	HW	Size of spaces through which ant can pass (Sarty, Abbott & Lester, 2006); mandibular musculature (Kaspari, 1993)	
Mandible Length	ML	Indicative of diet; longer mandibles could allow predation of larger prey (Fowler et al., 1991)	
Top tooth length	TT	May function to cut and masticate; longer top teeth may increase functional complexity, increasing ability to cut and break down plant material (Santana, Strait & Dumont, 2011)	
Scape Length	SL	Mechano and chemoreception (Schneider, 1964)	
Antenna Length	AL	Mechanoreception; length of antennae inhibits ability to sense surroundings (Schneider, 1964)	
Max. spine length	MSL	Spines may act as anti-predatory mechanisms (Michaud & Grant, 2003)	
Max. hair length (alitrunk)	MHL	Hairs may increase tolerance to dehydration (Cloudsley-Thompson, 1958) they may function in thermo-regulation (Heinrich, 1974) or relate to mechanoreception	
Mid-femur length	FEM	Indicative of foraging speed, which reflects habitat complexity (Feener, Lighton & Bartholomew, 1988)	
Mid-tibia length	TIB	Indicative of foraging speed, which reflects habitat complexity (Feener, Lighton & Bartholomew, 1988)	
Mid tarsus length	TAR	Linked to locomotion and climbing ability (Gladun & Gorb, 2007)	
			
Count measures			
Maxillary palp segments	MP	Liquid sugar feeding (Eisner, 1953)	
Labial palp segments	LP	Functions in ability to taste (Homberg, Christensen & Hildebrand, 1989)	
Spines (alitrunk)	ST	Spines may act as anti-predatory mechanisms (Michaud & Grant, 2003)	
Spines (petiole)	SP	Spines may act as anti-predatory mechanisms (Michaud & Grant, 2003)	
Number of Teeth	T	May increase ability to masticate prey or plants (Santana, Strait & Dumont, 2011)	

All trait variables were log-transformed before analysis. Weber’s length (distance from the anterodorsal margin of the pronotum to the posteroventral margin of the propodeum (Weber, 1938; Brown, 1953) was used as the main descriptor of body size (Diniz-Filho et al., 1994), while other traits were used to describe the shape. The principal morphological trait differentiating amongst species was body size. We therefore corrected for size by using residuals of linear regressions against Weber’s length as response variables (Kaspari & Weiser, 1999; Gibb & Parr, 2013) to characterize body shape relative to static allometry: the phenomenon that each body part scales with overall body size (Stern & Emlen, 1999) of each morphospecies (Gould, 1966).

To reduce the total number of traits measured, we tested for correlations amongst this set of 20 traits. Pairs of traits which were significantly correlated were examined (using a Spearman Rho for test of the significance of correlations) and we retained the trait we considered to be most functionally meaningful of the pair (Table S1).

Using this approach, a reduced list of ten traits were selected (Table 1, Table S1). These traits were then measured on up to six individuals of each morphospecies (fewer if we had fewer than six specimens). We used average trait values for all analyses, even for highly polymorphic genera (e.g., Camponotus and Melophorus). We did not include soldiers of dimorphic genera such as Pheidole, as soldiers were relatively rare. Measured specimens were taken from all three areas and both habitat types when possible. The number of maxillary palp segments was recorded from Australian Ants Online, and this trait was only recorded at the genus level (Shattuck & Barnett, 2001). All other traits were measured from the left-hand side of ants.

Data analysis

Morphospace characterisation

Ant genera and subfamilies were plotted in morphological trait space using Principal Components Analysis (PCA). Average trait values were calculated for each genus, which were then normalised within Primer 6® (Clarke & Gorley, 2006). PCA reduces the dimensionality of large multivariate data sets, by deriving variables (Principal components), which are linear combinations of the original variables. These principal components often retain most of the variability in the original traits (Jolliffe, 2005). The use of trait residuals with Weber’s lengths removed the effect of body size from trait measures, so normalised Weber’s lengths were also plotted against Principal components 1. We also plotted PCs 2 and 3, in order to encompass a greater percentage of the variation in our data set.

Phylogenetic relationships

To determine the strength of the underlying phylogenetic relationships in driving trait–environment interactions, we used the PDAP (Phenotypic Diversity Analysis Programs) module of Mesquite (Garland, Harvey & Ives, 1992) to create phylogenetically independent contrasts of the morphological traits and Weber’s length using Felsenstein’s method (Felsenstein, 1985). This method computed weighted differences between character values of pairs of sister morphospecies (and/or nodes), as indicated by a phylogenetic tree, and working down the tree from its tips, the procedure results in n−1 contrasts from n original tip morphospecies. Each of these contrasts is thus considered independent of the others in terms of the evolutionary changes that have occurred to produce differences between the two members of a single contrast (Garland, Harvey & Ives, 1992).

We assembled a phylogenetic tree for the ant genera sampled in the study using the most recent comprehensive phylogeny to genus (Moreau et al., 2006). Morphospecies within each genus were treated as unresolved soft polytomies at each individual genus tip, meaning we were adjusting for genus-level independence. Phylogenetically independent contrasts were standardized and regressed against Weber’s length to produce residuals using the protocol described by Garland, Harvey & Ives (1992). We used these new phylogenetically adjusted residuals in a Principal Components Analysis (PCA) to see whether the same trait associations were replicated. If there was a strong tendency for closely related genera to share morphologies, the phylogenetic independent contrasts analysis would give a different inference from the initial trait analysis. If there is little difference between phylogenetic independent contrasts and phylogeny-free trait contrasts, it would suggest that the patterns of trait association are similar, regardless of the inherent phylogenetic structure among genera.

The phylogenetic independent contrasts compared traits across morphospecies, but, we were also interested in whether the new compound PCA axes were independent of phylogeny for morphospecies within genera. We used Mantel tests to look for correlations between phylogenetic distance and morphological distance among all morphospecies pairs (Mantel, 1967; Peres-Neto & Jackson, 2001). A patristic distance matrix for all morphospecies pairs was produced from the phylogenetic tree (distances between species calculated whereby each branch length was considered to be equal to one unit, Desdevises et al., 2003). Separate morphological distance matrices were generated for log (body length) and each principal component axis using Euclidean distances. Mantel tests and calculations of morphological distances were performed using PC-ORD 6.0 (McCune & Mefford, 2011).

We removed the MP trait (number of maxillary palp segments) from all phylogenetic relatedness analyses, as this trait was categorical (1–6 maxillary palp segments) and not a continuous measurement.

Fourth corner

The problem of associating species traits and environmental variables using species abundance data is known as the fourth-corner problem (Legendre, Galzin & Harmelin-Vivien, 1997). This problem can be thought of as a 3 table problem which takes environmental data (R), species abundances (L), and species traits (Q) and uses these three tables to understand how traits associate to the environment (D). We used the fourth-corner modelling approach to assess the relationship between morphological traits and the environmental measures: microhabitat variables and macrohabitats, using the ‘trait.mod’ function developed by Brown et al. (2014) for R 2.15.1 (R Development Core Team, 2012).

For this analysis, we used the matrix of species abundances at each site (L), the matrix of microhabitat variables at each site (R) and the matrix of ant traits for each species (Q). The approach fits a predictive model for abundance of each ant species at each site (L) as a function of the microhabitat variables (R), morphological traits (Q) and the trait–environment interaction terms (the ‘fourth corner’). The environmental variables we used were tall grass cover, herb cover, bare ground cover, short grass cover, leaf litter cover, C:N, P (all microhabitat scale) and macrohabitats.

We selected a generalised linear model (Nelder & Wedderburn, 1972), with the negative binomial family for our fourth corner analysis because the count data (for species abundances) was overdispersed. The fourth corner approach uses LASSO for variable selection (Tibshirani, 1996) with an algorithm to allow the use of a negative binomial option (Brown et al., 2014). LASSO sets model terms that do not explain any variation to zero. Any interaction terms presented in the results that are zero mean that the variables do not interact in predicting abundances of ants.

This fourth-corner modelling method compliments the fourth-corner hypothesis testing (Legendre, Galzin & Harmelin-Vivien, 1997) by providing information not only on the associations between microhabitat variables and species traits, but also providing coefficients that quantify the strength of the associations. These associations indicate how the macrohabitats, microhabitat and morphological trait variables influence ant abundances. The resulting coefficients are expressed as positive or negative values: a positive association between a trait, e.g., Weber’s length, and bare ground cover indicates that there are more large ants in sites with a greater percentage cover of bare ground. Alternatively, a negative coefficient would indicate that ants were smaller when bare ground cover was higher. The absolute size of the coefficient indicates the strength of the association, while the sign (positive or negative) of the coefficient indicates the direction of the relationship. Habitat type is considered as a two level factor, so coefficients of remnants indicate the mean abundance of traits in comparison with pastures (e.g., a coefficient value of 1.5 Weber’s length in remnants means that the mean abundance of ants with long Weber’s lengths is log(1.5) more than the mean abundance of ants with long Weber’s lengths in pastures).

Trait variability within macrohabitats

A permutational multivariate analysis of dispersion (PERMDISP) for Primer 6® was run on a Euclidean distance matrix of each morphological trait amongst macrohabitats. PERMDISP was employed to test for differences in the homogeneity of ant trait measurements across habitats. The analysis compares the differences from observations to their group centroid (analogous to a measure of variance) and allows us to compare the heterogeneity of a trait between habitats (Anderson, Gorley & Clarke, 2008).

The dataset used for these analyses can be found at Figshare (Yates, Andrew & Gibb, 2013).

Variation in microhabitat variables between macrohabitats was tested but there was no significant differences found (ML Yates, unpublished data). These results were not included as they do not assist in the interpretation of the large dataset we have included.

Results

We measured traits on 302 individuals belonging to 123 morphospecies (see Table S2 for information on the individuals we used per site). There were many singletons, and while two subfamilies were represented by only one genus, most subfamilies and genera found were well represented within macrohabitats (Table S2).

In terms of genera and their positions in morpho-space, the residuals of the nine continuous traits (see Table 2), were reduced to four principal components. Principal Components 1, 2, 3 and 4 cumulatively accounted for 80.4% of the variance of genera in morphological trait space. Principal Component 1 (PC1) accounted for 33.7% of the variance, with a large negative coefficient for head length (Table 2). Genera with negative PC1 loadings had shorter heads, relative to body length, e.g., Anillomyrma and Meranoplus (Fig. 1A). Principal Component 2 (PC 2) described 21.4% of the variance, with large negative coefficients for mandible and top tooth length, and positive coefficients for scape length (Table 2), e.g., Myrmecia appeared low on this axis, while Aphaenogaster and Strumigenys appeared high (Fig. 1B). Principal Component 3 (PC 3) described 14.9% of the variance, with large negative coefficients for scape and femur length (Table 2). Amblyopone and Cerapachys were positioned high on this axis, having the shortest scapes and femurs, relative to Weber’s length (Fig. 1B). Lastly, Principal Component 4 (PC 4) accounted for 10.3% of variance, with a large positive coefficient for maximum hair lengths on the thorax and a large negative coefficient for eye width (Table 2). Those genera with large positive coefficients had longer maximum hairs on the altitrunk, e.g., Cerapachys, and those genera with large negative coefficients have narrower eyes, e.g, Strumigenys.

Figure 1 Ant genera plotted in morpho-space: (A) Weber’s length against PC1; (B) PC2 against PC3.

Morphospecies distinguishable in subfamilies by numbers 1–9. 1. Myrmicinae (Adl, Adlerzia; Ani, Anillomyrma; Aph, Aphaenogaster; Car, Cardiocondyla; Cre, Crematogaster; Epo, Epopostruma; Mayr, Mayriella; Mer, Meranoplus; Mon, Monomorium; Phe, Pheidole; Sol, Solenopsis; Str, Strumigenys; Tet, Tetramorium); 2. Amblyoponinae (Amb, Amblyopone); 3. Formicinae (Sti, Stigmacros; Pol, Polyrhachis; Pro, Prolasius; Cam, Camponotus; Mel, Melophorus; Not, Notoncus; Opi, Opisthopsis; Nyl, Nylanderia; Pla, Plagiolepis); 4. Cerapachyinae (Cer, Cerapachys); 5. Dolichoderinae (Tap, Tapinoma; Tec, Technomyrmex; Iri, Iridomyrmex; Dol, Dolichoderus; Och, Ochetellus); 6. Heteroponerinae (Het, Heteroponera); 7. Ponerinae (Hyp, Hypoponera; Lep, Leptogenys; Pac, Pachycondyla; Plat, Platythyrea); 8. Myrmeciinae (Myr, Myrmecia); 9. Ectatomminae (Rhy, Rhytidoponera).

Table 2 Results of Principal Components Analysis (PCA) on average trait values for genera.

All traits except Weber’s length are based on residuals with Weber’s length. Principal Components for each trait are represented as eigenvectors. PC1, PC2, PC3 and PC4 together account for 80.4% of the variation in morphological traits. Traits contributing to more than 30% of variation in first two Principal Components are shown in bold (maxillary palps trait removed as this is a categorical trait, not continuous).

	Principal components contribution	
% Variation	33.7	21.4	14.9	10.3	
Trait	PC1	PC2	PC3	PC4	
Weber’s length	0.365	−0.108	−0.402	−0.064	
Min inter-eye distance	−0.399	−0.121	0.28	−0.281	
Eye width	−0.27	−0.301	0.168	−0.571	
Head length	−0.429	0.253	0.176	−0.059	
Mandible length	−0.335	−0.424	−0.381	0.16	
Top tooth	−0.295	−0.473	−0.38	0.151	
Scape length	−0.188	0.495	−0.42	−0.302	
Max hair length on thorax	−0.317	0.109	0.254	0.667	
Femur length	−0.341	0.397	−0.412	0.045	

The PCA conducted on the residuals of phylogenetic independent contrasts produced axes with very similar trait associations to the PCA of uncorrected trait residuals for most traits (Table 3). This indicates that the trait associations apparent are general, regardless of the inherent phylogenetic structure among genera. Furthermore, Mantel tests revealed only two of the continuous morphological traits we measured (Weber’s length and Scape length) were significantly related to phylogeny (Table 3).

Table 3 Results of Mantel tests and summation of principal components analysis performed

(A) residuals of phylogenetically independent contrasts (PICs), and (B), residuals of log-transformed traits regressed against log body length showing the coefficients for each trait and the variation explained by each principle component (PC1, PC2 and PC3). Mantel tests relate a phylogenetic distance matrix to a distance matrix based on each of the respective continuous trait distance matrices. The traits which are significantly correlated to phylogenetic relationships are in bold, and those coefficients which are high are in bold (maxillary palps trait removed as this was a categorical trait).

	Mantel test	(A) PICs	(B) Trait residuals	
Trait	r	t	p	PC1	PC2	PC3	PC1	PC2	PC3	
Weber’s Length	0.08	2.48	0.01	−0.45	0.23	0.02	0.37	−0.11	−0.40	
Min inter-eye distance	0.04	0.96	0.34	−0.15	−0.12	−0.05	−0.40	−0.12	0.28	
Eye width	0.06	1.8	0.07	−0.39	−0.16	0.18	−0.27	−0.30	0.17	
Head length	0.01	0.18	0.86	−0.34	−0.48	0.64	−0.43	0.25	0.18	
Mandible length	0.04	0.89	0.37	−0.34	−0.11	−0.04	−0.34	−0.42	−0.38	
Top tooth	0.02	0.51	0.61	−0.10	−0.03	−0.05	−0.30	−0.47	−0.38	
Scape length	0.08	2.35	0.02	−0.41	0.36	−0.19	−0.19	0.50	−0.42	
Max hair length on altitrunk	0.04	0.84	0.4	−0.15	−0.66	−0.69	−0.32	0.11	0.25	
Femur length	0.07	1.88	0.06	−0.43	0.31	−0.20	−0.34	0.397	−0.412	
										
Variation (%)				95.7	1.7	0.8	33.7	21.4	14.9	

Is the pattern in morphological traits independent of phylogenetic relationships?

There was a significant correlation between phylogenetic distance and distance in scape length, and Weber’s length (Table 3). So the phylogenetic distance between these species is correlated to the distance between species in terms of their respective scape lengths. This implies that, for these two continuous morphological traits, we could not separate a phylogenetic effect from ecological effect. Appendage size was not correlated with phylogenetic distance, suggesting that it is determined more by the environment.

Do morphological traits respond to the environmental variables: macrohabitat and microhabitat?

The fourth corner model performed well, explaining 39% of variation with a 5.5% standard error. Only macrohabitats showed significant associations with morphological traits (Fig. 2). Microhabitats variables were not associated with any morphological traits. One morphological trait, minimum inter eye-distance, was the only trait which was not associated with macrohabitats.

Figure 2 A graphical representation of the 4th corner interaction coefficients for the abundance model.

Significant associations are shown in blue or red, and the relative tone of colour indicates the strength of association. Red represents a positive association, blue represents a negative association. The coefficient values are expressed on a log scale. Along the x axis are the environmental variables (microhabitat variables and macrohabitat): CN, C:N; P, phosphorus; Tall Grass, tall grass cover; ShortGrass, short grass cover; Herb, herb cover; LeafLitter, leaf litter cover; Bare Ground, bareground cover; Average temp, Average ambient daily temperature (°C); HabitatRT, Remnant habitat type (coefficients of remnants indicate what the mean abundance of traits are in comparison with pastures).

Ants in remnants had higher counts of maxillary palp segments, longer scapes, wider eyes and longer mandibles than those in pastures. The mean abundance of ants with long Weber’s lengths, head lengths, top teeth, maximum hair lengths on thorax, and femur lengths, were significantly lower in remnants, compared to the overall means of these traits (Fig. 2). This means that ants in remnants have shorter bodies, shorter heads, shorter top teeth, shorter maximum thorax hairs and also shorter femurs, than average.

Do morphological traits vary more in remnant macrohabitats?

Head length and top tooth length were the only traits out of the nine associated traits whose variability significantly differed between habitat type (F = 6.4712, P (perm) = 0.01; F = 7.01, P (perm) = 0.02, respectively). Average head length was higher and more variable in remnants (Pasture = 0.41 mm, ±0.09 mm; Remnant = 1.03 mm, ±0.22 mm). Top tooth length of ants was on average longer and more variable in pastures (Pastures = 1.06 mm, ±0.24 mm; Remnant = 0.36 mm, ±0.08 mm).

Discussion

This study is the first of its kind to use a multi-scalar approach to explore the relationship between morphological traits of organisms and their environment. Although morphological and behavioural traits have previously been shown to respond strongly to fine-scale habitat features, such as structural complexity (Farji-Brener, Barrantes & Ruggiero, 2004; Gibb & Parr, 2010), we detected relationships only at site scales. Differences in the traits of morphospecies were pronounced between the two macrohabitats (pastures and remnants). We also found evidence that most of the traits we measured were independent of phylogeny at both genus and morphospecies level (which was unexpected considering other work which show traits are dependent upon phylogenetic relationships (see Silva & Brandão, 2010; Barton et al., 2011). Our findings thus suggest that trait–environment relationships are strongly dependent on scale.

Macrohabitats

Fourth corner analysis showed clear morphological differences between assemblages of ants at macrohabitat scales. Many key traits drove these differences, in particular, the number of palp segments, scape length and eye size were greater in remnants, while relative leg length, body size and apical mandibular tooth size were greater in pastures. Two traits showed strong phylogenetic dependence, but this does not necessarily indicate that the relationship with habitat does not relate to function. Rather, it suggests that it should be interpreted with greater caution, due to the risk of confounding with other phylogeny-related traits. Although we did not detect strong microhabitat-trait relationships, several of these relationships are likely to be a result of structural differences between remnants and pastures (Dormann, 2007). In agreement with previous studies (Kaspari, 1993; Kaspari & Weiser, 1999; Farji-Brener, Barrantes & Ruggiero, 2004; Ness & Bronstein, 2004; Sarty, Abbott & Lester, 2006; Cunningham & Murray, 2007; Gibb & Parr, 2013), morphospecies living in structurally simple habitats are larger, with relatively long legs and heads. In complex habitats, such as in the dense litter in remnants, invertebrates occupy the interstices between litter and the soil, and large body size and relatively long legs are both impediments to movement (Kaspari & Weiser, 1999). Additionally, ants commonly found in complex habitats do not walk long distances (Silva & Brandão, 2010), so ants found in pastures may have longer legs as a result of further foraging distances to food.

Relationships between feeding morphology and macrohabitat are not well documented for ground dwelling invertebrates, but greater abundance of ants with longer mandibular teeth detected in pastures suggests that these macrohabitats may favour predatory morphospecies. This finding is consistent with recent work on stable isotope signatures of ants, showing that assemblages become less predatory when pastures are restored to a more remnant-like state (Gibb & Cunningham, 2013). But this hypothesis is complicated by our mandible findings; that ants in remnants have longer mandibles.

There is an alternative explanation for these feeding traits. Firstly, there may be larger prey in remnants as a result of increased macrohabitat complexity (where longer mandibles for attacking larger prey would be important (Fowler et al., 1991)), whilst ants in pastures may have more herbivorous diets (as a result of reduced macrohabitat complexity), hence having longer mandibular teeth for mastication may be more valuable (Santana, Strait & Dumont, 2011). The significant variation in top tooth length of ants in pastures suggests also that there is greater variety of food which requires mastication within pastures. Granivory is a function which is common in more open, savannah like habitats (Andersen & Lonsdale, 1990) and mandibular teeth may have a significant role in masticating plant seeds (Table 1). Larger bodies and longer top teeth may both be associated with pastures, as a result of more granivorous feeding habitats (Kaspari, 1996; Ness & Bronstein, 2004).

Positive associations between occupation of remnants and eye width, scape length and the number of maxillary palp segments suggest that sensory morphology is particularly important in these macrohabitats. These traits are all related to an ant’s ability, to navigate, sense and move through its surroundings (Table 1), signifying that remnants are more perceptually demanding, possibly because they are more complex and heterogeneous. Maxillary palp segments and scape length may be associated with liquid sugar feeding (Jervis, 1998), so this may reflect the greater occurrence of sugar-dependent morphospecies in remnants, where trees provide both nectar and honeydew (Gibb & Cunningham, 2013). Maxillary palp counts are greater for Formicine and Dolichoderine ants than for other subfamilies (ranging from 50% as many more mandible palp counts, to 300% more), so this trait was clearly associated with phylogeny; liquid sugar feeding is also strongly associated with these subfamilies (Davidson, 1997; Cook & Davidson, 2006). Thus, although phylogeny determines many of these sensory traits, there is a clear logical link between the function of the traits and the environment.

Only one trait was not associated with macrohabitats. It was surprising that minimum eye distance, which is an indicator of habitat complexity (Gibb & Parr, 2013), was not significantly associated with macrohabitats. We expected ants within remnants to have eyes further apart based on the supposition that complex habitats require organisms to be more capable of seeing obstacles around them.

Microhabitat variables

Small-scale microhabitat factors are generally considered important in driving assemblages (Suggitt et al., 2011) and previous studies on ant assemblages have revealed strong trait–environment relationships at small scales (Kaspari, 1993; Gibb & Parr, 2010; Silva & Brandão, 2010). It was therefore surprising that our analyses revealed no significant relationships between microhabitat variables and morphological traits. But, we may not have measured the specific variables which are important drivers of morphological traits (for example clay content of soil, soil texture and soil strength have been found to be an important driver of ant assemblage structure (Bestelmeyer & Wiens, 2001; Debuse, King & House, 2007).

Understanding environmental influences on assemblages is confounded by effects of scale (Sanders et al., 2007). The scale in which we measured microhabitat variables (plot scale), may have been too coarse to be influencing on ants (Suggitt et al., 2011), and our plots may not have accounted for enough variation in the microhabitats that constitute each macrohabitat. Alternatively, the variation of microhabitat variables within macrohabitats may have been too high to be easily associated with morphological trait variation. We may have also encountered autocorrelation through such artefacts as grouping plants into herbs and grass cover, as opposed to recording individual plant species (Dormann, 2007). Our sampling within each area may have also underrepresented landscape scale dynamics, which are an important influence on within habitat patch populations (Spiesman & Cumming, 2008). Nevertheless although our study has some limitations, we have found clear evidence that invertebrate functional traits respond strongly to macro-scales.

Synthesis

Almost all morphological traits were associated with macrohabitats. This was in direct contrast to our study assessing ant community structure (the current study is a subset of this data) which found strong associations of community structure to microhabitat variables such as C:N, phosphorus, herb and leaf litter cover (Yates, Gibb & Andrew, 2012). The unique combinations of microhabitat variables which characterise the macrohabitats may, therefore, be more important for traits than any single specific microhabitat variable, but this does not annul the importance of unravelling small scale microhabitat influences in ecology (Suggitt et al., 2011). Even so, only two traits were linked to phylogeny, which suggests phylogenetic diversity alone may not encapsulate functional responses to the environment (Srivastava et al., 2012).

Although several trait–environment associations are linked to phylogeny, these associations are in accordance with the predictions that traits would respond to macrohabitats, and are likely to be meaningful. Interpretation is clearer for phylogenetically independent traits such as leg length, eye size and mandible morphology, which show distinct links to macrohabitats. Although morphological measures can be arduous to attain, they provide the potential to extend the understanding of the interactions between species and their environment beyond localised or even continental faunas. Morphological traits are directly related to the interaction between a species and its environment, and, in this study, relationships are easily linked to knowledge of the functional ecology of species associated with specific traits. In addition, traits may be indicative of the importance of biotic structuring mechanisms, such as competition (Nipperess & Beattie, 2004). We highlight the scale-dependence of trait–environment relationships and advocate greater use of this method for detecting broad-scale generalities in community ecology.

Supplemental Information

Figure S1 Map of climatic gradient and paired habitats which were assessed

Circle outlines encompass the three individual zones while solid circles represent the three paired sites within each zone. Squares represent the main town of the zone where the following climatic data was collect from (Average climatic data for the past 15 years and elevation for Boomi, Inverell and Armidale: Mean minimum temperature (°C): Boomi = 12.9, Inverell = 7.4, Armidale = 7.1; Mean maximum temperature (°C): 27.6, 23.9, 20.3; Average rainfall (mm/yr): 505.9, 805.3, 791.5; Elevation (m.a.s.l.) = 160, 582, 980) (BOM 2010).

Click here for additional data file.

Table S1 Correlation coefficients amongst ant morphological traits (n = 24 genera). Significant correlations (p < 0.05) are shown in bold.

Click here for additional data file.

Table S2 Number of individuals measured from each morphospecies, within subfamilies, in each macrohabitat (R = remnant, P = pasture).

Click here for additional data file.

Thank you to all of the landholders who kindly allowed sampling to be conducted on their farms. Many thanks also to Dr. Saul Cunningham (CSIRO), and Dr. Philip Barton (ANU) for their helpful comments on a previous version of this manuscript.

Additional Information and Declarations

Competing Interests

Author Contributions

Data Deposition

NRA is an Academic Editor for PeerJ. There are no other competing interests.

Michelle L. Yates conceived and designed the experiments, performed the experiments, analyzed the data, wrote the paper.

Nigel R. Andrew conceived and designed the experiments, contributed reagents/materials/analysis tools, wrote the paper.

Matthew Binns analyzed the data, wrote the paper.

Heloise Gibb conceived and designed the experiments, analyzed the data, contributed reagents/materials/analysis tools, wrote the paper.

The following information was supplied regarding the deposition of related data:

Yates M, Andrew NR & Gibb H (2013) Dataset for Morphological traits: predictable responses to macrohabitats across a 300 km scale.

Figshare DOI: http://dx.doi.org/10.6084/m9.figshare.878459.

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
