# Peer review of "Morphological traits: predictable responses to macrohabitats across a 300 km scale"

_PeerJ, doi:10.7717/peerj.271_

## Round 0.1 · original submission · Minor Revisions

· Academic Editor

Minor Revisions

Both reviewers (and I) found the data you present interested and certainly merit publication. The reviewers both offer suggestions on some textual corrections which I don't think will be much of a problem for you. The more substantive issues raised are aspects of the analysis. I am largely at the mercy of the reviewers and you regarding analysis, as the techniques used here are not ones with which I have any experience. Consequently, please take a hard look at the comments/suggestions regarding the analyses, and if you think a suggested change is inappropriate please offer an explanation. I think the most significant issue pertains to the question regarding the nature of the data used in the PERMDISP analysis. As I understand it (and correct me if I'm wrong), the reviewer believes the distances used in the analysis are have been standardized in such a way as to make the PERMDISP analysis inappropriate. The suggested solution is to use PERMDISP on euclidean distances. Please make certain you respond to this issue, as I believe it is the only point in the reviews that suggests a flaw in the analysis (as opposed to suggestions for more informaiton or for better presentation).

On just a couple other points, we can do color in your graphs at PeerJ, and as the pdf of your paper will also include color, I think this option (as suggested in the reviews) would improve the clarity of some of your figures. I notice that both reviewers wanted to see data sets provided with the manuscript. This is not a requirement for publication at PeerJ, it is left to the discretion of the authors. However, we do have the capacity for including data with your publication, and it is encouraged.

Best of luck with your revision, and I'll be looking forward to seeing the revised manuscript. Naturally, if you have any questions please don't hesitate to contact me.

Reviewer 1 ·

Basic reporting

No comments

Experimental design

Line 111 “more primary” more primary according to what criterion?

Was there any signficant variation in microhabitat variables between macrohabitats? If so, that might be why you're seeing only an effect of macrohabitat in the 4th corner analysis.

line 143 "We therefore corrected for body size by using residuals of linear regression ..." were these regressions using log-transformed traits or original traits? Allometric relationships are usually power laws so this should be log-transformed.

4th corner analysis seems appropriate here, but the real contribution of that approach is that it deals with non-independence between rows of the matrix L by using permutations to calculate significance of the interaction terms. You seem to be using a modification of the approach described in Legendre etal 1997 that is described in Brown et al (in review). As a result I cannot judge the adequacy of these methods in the present paper. For example, which permutation model did you use, if any (Legendre et al have 4!)

Validity of the findings

I saw no indication that the data are or would be made available.

Line 287 "...explaining 39% of the variation with a 5.5% standard error."
Do you mean that the residual deviance of your Negative Binomial GLM was 39% as large as the residual deviance? What does the standard error refer to? Normally this is reported for a coefficient in the model?

Line 273-274 you conclude that the PCA axes done on the PICs are similar to the uncontrolled trait variation, but e.g. eye width loads positively when done with a PIC and negatively on PC1 when uncontrolled? How can you support this conclusion better -- maybe start by putting the PCA axes in the same table. Maybe they are ... section (b) of Table 3 has the same values but without the signs, in which case Table 2 is redundant.

Table 3 raises alot of questions -- what are the 2nd row of numbers under the mantel test statistics for each trait? Why are some of the loadings for the principle components divided up into 2 rows?

LInes 277 + 280 refer to table 2 but should be table 3.

Additional comments

Lines 207 + 208 delete, and move 209 and 210 to the end of 211 – 215.

·

Basic reporting

Basic reporting is fine except for a couple of things.

1. I may have missed it but I'm not seeing any mention of the base data being made available anywhere. PeerJ policy is that data should be shared. Morphometric means per species per site (or some reasonable summary of that) would be appropriate I think. See Appendix D of Nipperess & Beattie (2004) Ecology 85: 2728-2736 for an example (can be found here: http://esapubs.org/archive/ecol/E085/086/). Data could be included as a supplementary material in PeerJ or maybe submit to DataDryad.

2. Figures 1 and 2 are poor resolution, making them hard to read. In the case of figure 1, I suggest re-plotting the PCA scores in Excel (or something similar) and saving the figure as PDF. The current labelling system for figure 1 could be improved as well. I'm pretty sure that colour isn't an issue with PeerJ, so perhaps you could have a different colour for each subfamily, which is easy enough to do in Excel.

3. I suggest a couple of corrections to the text as follows:

Line 115: the terminology gets a little confusing. Maybe say: "pitfall traps were set up in sets of five at the three haphazard plots" instead.
Line 156: "areas" not "areass"
Line 189: I think you have that backwards. The PIC analysis removes the effect of phylogeny, so the initial (without PIC) trait analysis would not be "phylogeny-free" but would include the effects of phylogeny.
Line 280: Table 3 not table 2
Line 344: Word choice error. Use "herbivorous" rather than "vegetative". While "vegetative" can mean simply something related to vegetation, it usually refers to either mode of propagation or the state of being comatose.

Experimental design

Analyses were conducted differently to how I would have done them but, in general, I have no problem with either the design or the methods of analysis, with one exception.

The exception is the use of PERMDISP to test for differences in the homogeneity of trait measurements amongst macrohabitats (lines 246-251). This is a multivariate method that was applied to univariate data (one trait at a time). Used in this way, the test compares differences in observations from their group mean, rather than the centroid (which is the multivariate equivalent of a mean). Permutation is then used to test whether dispersion about group means is significantly different between groups. That usage is fine - it's a kind of non-parametric test of homogeneity. The problem is that the distances of each observation from the group mean was based on Bray-Curtis Dissimilarity. This measure is intended for count data (e.g. species abundances) and includes a standardisation that makes no sense for morphometric data. I advise that you re-do these analyses using euclidean distance. Then the differences are simply the absolute differences in the trait value of the observations from their respective group means, and PERMDISP is a simple univariate and non-parametric test of those differences between groups. I'm pretty sure that PRIMER will give you the option of running PERMDISP on euclidean distances. If not, Marti Anderson has a stand-alone version of PERMDISP on her website (https://www.stat.auckland.ac.nz/~mja/Programs.htm) that allows for euclidean distances.

The fourth corner analysis is very interesting. I look forward to the forthcoming paper by Brown et al.

Validity of the findings

As I said previously, that data will need to be publicly available in some form to satisfy PeerJ's policy.

Conclusions are sound and based on the data, although interpretation may change slightly when the PERMDISP analysis is redone.

I am confused by the statement on lines 384-386. How is your grouping of plant species related to spatial autocorrelation? I don't understand the reasoning behind that statement.

Additional comments

Interesting paper on the relationship between morphology and environment. A useful contribution to an emerging field in ecology.

---

## Round 0.2 · accepted · Accept

· Academic Editor

Accept

Dr. Andrew (and colleagues), thanks so much for the very thorough review and helpful letter documenting your responses to the reviewers questions. I think the revisions strengthen your interesting paper. I do want to apologize for my delay in accepting your manuscript. The delay is entirely my fault, a combination of the start of the semester and various family issues. In any event, I am confident that the Peer J staff will move forward expeditiously in processing your manuscript.